# NOD2 Signaling Circuitry during Allergen Sensitization Does Not Worsen Experimental Neutrophilic Asthma but Promotes a Th2/Th17 Profile in Asthma Patients but Not Healthy Subjects

**DOI:** 10.3390/ijms231911894

**Published:** 2022-10-06

**Authors:** Mélodie Bouté, Saliha Ait Yahia, Ying Fan, Daniel Alvarez-Simon, Han Vorng, Joanne Balsamelli, Julie Nanou, Patricia de Nadai, Cécile Chenivesse, Anne Tsicopoulos

**Affiliations:** Univ. Lille, CNRS, Inserm, CHU Lille, Institut Pasteur de Lille, U1019-UMR9017-CIIL-Centre d’Infection et d’Immunité de Lille, F-59000 Lille, France

**Keywords:** MDP, asthma, vaccine, NOD2, adjuvant, Th2/Th17-type inflammation, iBALT, allergy, airway remodeling

## Abstract

Nucleotide-binding oligomerization domain 2 (NOD2) recognizes pathogens associated with the development of asthma. Moreover, NOD2 adjuvants are used in vaccine design to boost immune responses. Muramyl di-peptide (MDP) is a NOD2 ligand, which is able to promote Th2/Th17 responses. Furthermore, polymorphisms of the NOD2 receptor are associated with allergy and asthma development. This study aimed to evaluate if MDP given as an adjuvant during allergen sensitization may worsen the development of Th2/Th17 responses. We used a mouse model of Th2/Th17-type allergic neutrophil airway inflammation (AAI) to dog allergen, with in vitro polarization of human naive T cells by dendritic cells (DC) from healthy and dog-allergic asthma subjects. In the mouse model, intranasal co-administration of MDP did not modify the AAI parameters, including Th2/Th17-type lung inflammation. In humans, MDP co-stimulation of allergen-primed DC did not change the polarization profile of T cells in healthy subjects but elicited a Th2/Th17 profile in asthma subjects, as compared with MDP alone. These results support the idea that NOD2 may not be involved in the infection-related development of asthma and that, while care has to be taken in asthma patients, NOD2 adjuvants might be used in non-sensitized individuals.

## 1. Introduction

Asthma is a chronic lung disease, affecting over 330 million people, with an estimated number of 420,000 people in the world that died from asthma in 2016 [1,2]. This disease is characterized by recurring symptoms of airway obstruction with airway hyperresponsiveness (AHR), bronchial remodeling, and an inflammatory process leading to the recruitment of eosinophils and/or neutrophils in the airways. Among asthma patients, up to 10% suffer from severe asthma. These asthmatics require a high-dose pharmacological therapy, i.e., inhaled corticosteroids combined with a second controller therapy [3], and display type 2 (T2) high and T2 low phenotypes [4,5]. In particular, the T2 low phenotype is often associated with a Th17-type cytokine profile, which participates in airway neutrophil infiltration [6,7], AHR [8,9], bronchial remodeling [9], and corticosteroid resistance [6]. Many different drivers of Th2/Th17 immune responses have been alleged in the development of asthma, including lung infectious diseases in infancy [10,11]. However, the involved molecular culprits remain elusive.

Pattern recognition receptors (PRRs) are key components for host recognition of danger signals and microorganisms [12] that ultimately shape the adaptive immune responses and, as such, are being used for vaccine adjuvants [13]. Among these adjuvants, ligands of the NOD-like receptor (NLR) NOD2 have been widely used. NOD2 senses both Gram-positive and Gram-negative bacteria, is mostly expressed by epithelial cells and dendritic cells (DC) and by other immune cells such as lymphocytes and macrophages [14,15]. NOD2’s main ligand, muramyl di-peptide (MDP), was recognized early as the minimal effective component of Complete Freund’s Adjuvant (CFA) (inactivated *Mycobacteria* in a water-in-oil emulsion), which is characterized as a general immunostimulant [16,17], eliciting both humoral and cellular responses [18,19]. NOD2 agonists have been used as efficient mucosal adjuvants in infectious diseases. As such, a TLR7/NOD2 agonist used as an adjuvant with HIV-1 p24 antigen induces mucosal protection in virus-infected mice [20]. An intranasal (i.n.) vaccination with the NOD2 agonist murabutide, a synthetic MDP derivative, and Norwalk virus-like particles increases anti-viral humoral and mucosal responses [21]. Moreover, co-injection of an encapsulated NOD2 ligand with biodegradable nanocarriers carrying HIV-1 Gag p24 antigen leads to an increased specific antibody response in comparison to alum, in mice [22,23]. On the mechanistic level, NOD2 enhances, through DC, IL-17 production by T cells [24] and directs IL-17 response during viral infection caused by *Citrobacter* and *Salmonella* [25]. MDP also acts as an adjuvant to the adaptive immune Th2 response, during systemic allergen immunization with ovalbumin (OVA) [26]. It is of interest that NOD2 recognizes *Streptococcus pneumoniae* [27] as well as viruses such as the respiratory syncytial virus [28], which are two childhood infections increasing the risk of developing asthma in adulthood [10,11]. Furthermore, polymorphisms of this receptor are associated with asthma [29,30] and allergy [31,32]. We have previously shown that NOD1, another receptor belonging to the NLR family, exacerbates type 2 allergic asthma in humans and mice, through its ligands, FK565 and FK156, respectively [33]. Altogether, these data suggest that, as is the case for NOD1, NOD2 ligands contained in pathogens or vaccines may potentially lead to increased susceptibility to asthma development or the worsening of asthma.

The aim of this work was to evaluate whether a NOD2 ligand given as an adjuvant during allergen sensitization may worsen subsequent Th2/Th17 responses. We used a murine model of dog-allergen-induced allergic asthma, exhibiting Th2/Th17-type immune responses. Intranasal NOD2 ligand co-administration during the sensitization phase did not alter the lung cytokine profile. Likewise, other asthma parameters including AHR, bronchoalveolar lavage (BAL) neutrophil recruitment, induced Bronchus-associated lymphoid tissue (iBALT) formation, and bronchial remodeling were not modified. The effect of co-stimulation was also evaluated in human nonasthmatic and dog-allergic asthmatic subjects in a model of DC co-cultured with naive T cells. As in mice, MDP/dog-allergen-primed DC from healthy subjects did not modify the T helper cell cytokine profile. Nonetheless, MDP/dog-allergen-primed DC from asthma patients elicited a Th2/Th17 profile, as compared with MDP alone, without additional effect on dog-allergen stimulation. These results suggest that the NOD2 pathway may not be responsible for the asthma-prone effect of some pathogens sensed by NOD2. Moreover, as a vaccine adjuvant, our data support the idea of a lack of inflammatory effects in healthy subjects but not in asthma patients.

## 2. Results

### 2.1. NOD2 Ligand Co-Administration Does Not Aggravate Allergen-Induced Airway Inflammation, Humoral Response, and AHR

To evaluate whether the NOD2 ligand may worsen Th2/Th17-type asthma through an adjuvant effect, we used a mouse model of dog-allergen-induced AAI, previously developed in our group [34]. The protocol is shown in Figure 1a and consists of an i.n. sensitization phase with allergen with or without MDP for the sensitized groups and PBS or MDP for the control groups. Seven days later, mice are challenged either with allergen for the two positive groups, or with PBS for the two control groups. As previously described, dog-allergen-challenged mice exhibited higher airway cell recruitment in the BAL than PBS mice. This elevated number of total cells was accounted for by the high numbers of neutrophils and also of some eosinophils, as well as by the increased lymphocyte numbers compared to the PBS control group (Figure 1b). As expected, exposure to dog allergen induced increased total and dog-allergen-specific IgE in sera (Figure 1c). Airway resistances to increasing doses of methacholine were measured to evaluate the functionality of this response. AHR was significantly increased in dog-allergen-challenged mice compared with PBS controls (Figure 1d). The NOD2 agonist MDP was administered during the sensitization phase, either alone or together with dog allergen. No significant modification in either of the above parameters was observed when mice were co-administered with dog allergen and MDP, compared with dog allergen alone (Figure 1b–d). These data indicate that this dog-allergen-induced chronic model exhibits a predominant neutrophilic airway inflammation associated with increased AHR that is not exacerbated by i.n. NOD2 co-administration.

### 2.2. Mixed Th2/Th17-Type Endotypic Profile Induced by Dog Allergen Remains Unchanged under the Effect of NOD2 Ligand

As NOD2 has been previously described to favor of a Th2 and Th17 profile [24,26], we evaluated whether the NOD2 ligand may influence the cytokine content of lung extracts in the different groups of mice. Although BAL eosinophil counts were low in this model, Th2-type cytokines *Il-13*, *Il-4*, and IL-33 were upregulated in the lungs of dog-allergen-sensitized mice, compared with the PBS group (Figure 2a). The lung Th17-type cytokines, *Il-17* and *Il-22*, were elevated after sensitization with dog allergen alone (Figure 2b), whereas the Th1 response was not modified (Figure 2c). However, lung Th2-, Th17-, and Th1-type profiles were not modified by MDP co-administration, as no difference was observed in the cytokine levels between the dog and dog/MDP groups (Figure 2a–c). We have previously shown, in the model of dog-allergen-induced AAI, that the main lung cells producing IL-17 were CD4^+^ T cells and γδ T cells [34]. To evaluate if the production of IL-17 was different among lung cells after MDP co-administration, flow cytometry analysis of IL-17 intracellular staining was performed in these two cell subsets. There was no difference in the expression of IL-17 in CD4^+^ and γδ T cells between the dog and the dog/MDP groups (Figure 3).

The production of the pro-Th2 chemokine CCL17, as well as of neutrophil-attracting chemokines such as CXCL1 and CXCL2, was also evaluated. Although the production of these chemokines in the lung was increased following sensitization with dog allergen alone, co-stimulation with MDP did not modify their levels (Figure 4). Altogether, these data suggest that when used during the sensitization phase, MDP co-administration does not increase the cytokine and chemokine profiles of the effector phase in this chronic model of asthma.

### 2.3. Dog-Allergen-Induced AAI Is Associated with Bronchial Remodeling without Additional Effect of NOD2 Ligand

Among the parameters of asthma, bronchial remodeling is a feature generally associated with the severity of the disease. Therefore, to evaluate if the NOD2 pathway may participate in this feature, a histopathological analysis was performed. Bronchial remodeling is characterized by increased mucus production, increased collagen deposition, and smooth muscle cells (SMC) hyperplasia. Histopathological examination of the lung sections from dog-allergen-challenged mice with or without MDP showed increased inflammatory infiltrates and mucus production, using periodic acid–Schiff (PAS) staining (Figure 5a,d,e), compared with controls groups (PBS or MDP). Peribronchial collagen deposits were assessed by Masson’s trichrome staining. Again, lung sections from dog-allergen-sensitized mice with or without MDP exhibited a significant increase in collagen fibers around the bronchi (Figure 5b,f). The peribronchial collagen was also examined by Picrosirius red staining, and the same results were observed. Finally, thicker bronchial SMC were observed in the positive groups compared with the control groups, by immunohistochemistry using an anti α-smooth muscle actin (SMA) antibody (Figure 5c,g). However, co-administration with MDP did not modify the inflammation, mucus production, and collagen deposition compared to dog-allergen sensitization alone (Figure 5a,b,d–f). For α-SMA staining, the co-administration did not change the thickness of SMC compared to dog-allergen sensitization alone (Figure 5c,g). To sum up, these data show that bronchial remodeling was not majored by MDP co-administration, which is an important issue in asthma.

### 2.4. NOD2 Ligand and Allergen Co-Administration Does Not Change the Formation of iBALT in Mice

The mouse model of dog-allergen-induced AAI leads to the formation of tertiary lymphoid organs, i.e., iBALT, composed of B, T cells, germinal centers with follicular dendritic cells (FDC) [34], and high endothelial venules allowing lymphocyte entry from the blood. Although the presence of iBALT has been observed in bronchial biopsies from asthmatic patients [35], the protective or deleterious role of these follicles in asthma is not clearly defined [36]. To ensure that the NOD2 ligand did not modify the formation of these follicles after co-administration with dog allergen, histopathological examination of the lung tissue was performed. Mice sensitized with dog allergen with or without MDP developed peribronchial ectopic follicles similar in number and size (Figure 6a). The formation of iBALTs is characterized by antibody production, so we next assessed local Ig production. BAL total IgA and dog-allergen-specific IgG1 were strongly increased in dog-allergen-challenged groups, whatever the sensitization with or without the NOD2 ligand (Figure 6b). To evaluate if the cell composition of these ectopic follicles were modified by MDP co-stimulation, we performed immunohistochemistry to stain T (CD3^+^), B (B220^+^), and FDC (CD21^+^) cells. Whether mice were co-sensitized by MDP or not, these three cellular types were equally present in the positive groups both in terms of number per iBALT and per surface area (Figure 6c–e). These data show that MDP did not modify iBALT formation in terms of size, composition, or functionality. Altogether, the data suggest that MDP/allergen i.n. co-administration does not aggravate the features of asthma in mice. It is, thus, unlikely that the presence of NOD2 ligands in inhaled pathogens associated with the development of asthma [10,11] may explain this asthma trajectory.

### 2.5. NOD2 Ligand and Allergen Co-Stimulation of DC from Healthy Subjects Does Not Alter DC Cytokine and T Cell Polarization Profiles

To check if the above mouse findings were also true in humans, monocyte-derived DC from nonasthmatic subjects (NA) were cultured in the presence of dog allergen with or without MDP co-stimulation. The supernatants were evaluated for IL-1β, IL-6, IL-23, and TGF-β1 production, which are cytokines that are involved in the differentiation of Th17 cells [37]. Regardless of the stimulatory conditions, there were no changes in DC cytokine levels, except for dog-allergen stimulation, which induced a small increase in IL-6 compared with the medium (Figure 7a). To check the effects of DC stimulation on the polarization of T helper cell subsets, primed DC from NA subjects were co-cultured with naive CD4^+^ T cells. The Th1 cytokine IFN-γ, the Th2 cytokines IL-5 and IL-13, and the Th17 cytokines IL-22, IL-17A, and IL-17F were assessed in the co-culture supernatants. In NA subjects, there was no induction of cytokines, in all conditions, except for a trend towards increased IL-17F production for dog-primed DC/T cell co-cultures (Figure 7b). Altogether, these results show that, as in mice, allergen combined with MDP did not modify the T cell profile in NA subjects, suggesting that MDP is not pro-inflammatory in healthy subjects.

### 2.6. NOD2 Ligand and Allergen Co-Stimulation of DC from Allergic Asthmatic Patients Promotes a Th2/Th17 Profile

Dog allergen/MDP co-stimulation of DC from dog-allergen allergic asthmatic (AA) patients slightly increased IL-1β and strongly increased IL-6 productions, compared with MDP. However, no difference was observed between dog and dog/MDP stimulation (Figure 8a). Intriguingly, higher levels of TGF-β1 were observed in AA patients than in NA subjects, in all conditions (*p* < 0.001). Like in NA subjects, only IL-17F was induced in T cells co-cultured with dog-allergen-primed DC from AA patients. However, when compared with MDP, co-stimulated DC from AA patients elicited the production of both Th2 and Th17 cytokines by co-cultured T cells (Figure 8b). Again, there was no difference between dog and dog/MDP conditions. Consistent with their allergic status, higher levels of IL-13 (*p* < 0.01) and lower levels of IFN-γ (*p* < 0.01) were found in AA patients than in NA subjects. These data show that in AA subjects, MDP/dog co-stimulation leads to a Th2/Th17 polarization profile, compared with MDP alone, which has no additional effect on dog allergen stimulation.

## 3. Discussion

*Streptococcus pneumoniae* and syncytial respiratory virus childhood infections have been associated with the development of asthma [10,11], yet the underlying mechanism remains elusive. These pathogens are recognized by the NOD2 receptor [16,17], which is able to shape the immune adaptive response towards Th2 as well as Th17 responses [24,26]. It is unknown, though, if this pathway plays a role in the emergence of asthma. Additionally, NOD2 ligands are being tested as respiratory mucosal adjuvants in vaccine formulation against various pathogens, which may favor the development of asthma in conjunction with allergen exposure. We, therefore, used our recently developed Th2/Th17 mixed asthma model to evaluate if the NOD2 pathway may favor development of this disease. For this purpose, we administered the NOD2 ligand via the i.n. route during the sensitization phase. In co-administered mice, in terms of classical features of experimental asthma, there was no modification of AHR, humoral response, BAL neutrophil, or eosinophil recruitment or of the lung Th2/Th17-type cytokine profiles. Hitherto, few studies have been published on the effect of NOD2 in asthma models. Wong et al. reported that in a model of OVA/alum-induced asthma, the intravenous administration of the NOD2 ligand MDP promoted BAL eosinophils, seric total IgE, and IL-13 but not IL5. However, the NOD2 ligand was not delivered at a mucosal site nor during the sensitization phase, as in our study, but, rather, it was delivered intravenously and during the allergen-challenge phase [38]. Another study by Duan et al. showed that the Th2 response could be restored in a model of OVA-induced respiratory tolerance, when the NOD2 ligand MDP was delivered intranasally prior to the sensitization phase [39]. Since the murine strain and the MDP dose administered were in complete concordance, the differences between our results and those of the aforementioned study might be related to the allergen, the model design, or the chosen timing of MDP administration, i.e., the sensitization phase in our study. In this context, we have recently shown that, despite the existence of MDP in house dust mite (HDM) extracts, the lack of NOD2 hindered eosinophil BAL cell recruitment but did not alter the lung cytokine profile, the humoral response, or the AHR in a model of HDM-induced asthma exhibiting a predominant Th2 profile [40]. In contrast, NOD1, as well as RIPK2 adaptor deficiency, strongly downregulated all the features of HDM-induced asthma [40]. Given that RIPK2 adaptor is a molecule common to both NOD1 and NOD2 downstream signaling, this suggests that there may be an upstream defect at the level of lung NOD2 activation in the dog-allergen model. We, thus, assessed the expression of *Pept1*, a specific transporter for NOD2 in the lungs of dog-allergen-challenged mice. There was no *Pept1* induction expression in this model suggesting the possibility that MDP is not transported across the epithelium to the intracellular NOD2 receptor through this pathway. However, in Duan ‘s paper, there was an effect of MDP delivered intranasally with an LPS-free OVA. Incidentally, LPS, commonly present in both HDM and dog allergen, has been shown to inhibit *Pept1* expression in mucosal intestinal tissue [41], putatively explaining the lack of effect of MDP in the dog model. One feature of severe asthma, not often thoroughly examined in animal models, is airway remodeling. Since NOD2 expression is increased in human bronchial SMC from asthma patients and MDP promotes their proliferation and migration [42], the effect of MDP co-administration on airway remodeling was examined in the dog-allergen model. There was no modification of the lung tissue inflammatory score, mucus production, or collagen deposition. It has been reported that NOD2 is required for the generation of the T helper follicular and plasma cells (two components of tertiary lymphoid tissues), after i.n. immunization with human serum albumin and cholera toxin [43]. Nevertheless, MDP/antigen co-stimulation without Cholera toxin was not able to induce a specific antibody response [43]. Consistently, together with the lack of additional effects of MDP on the humoral and cellular responses in the dog-allergen model, the composition and functionality of iBALT were not modified. Altogether these data suggest that i.n. allergen/MDP co-administration in naive animals does not aggravate the features of experimental asthma in this model and does not support its participation in the pro-asthma effect described for inhaled pathogens such as *Streptococcus pneumoniae* and respiratory syncytial virus.

To determine whether such findings held true in humans, we evaluated the effect of MDP/allergen co-stimulation in the DC obtained from NA subjects. Dog allergen induced a small increase in the production of IL-6, which may be accounted for by the small quantity of LPS present in the extracts (i.e., 2.5 ng). Indeed, the induction of IL-6 by LPS-stimulated monocyte-derived DC is a long-known feature that can be achieved with a concentration of LPS as low as 10 ng/mL [44]. The stimulation of DC from NA subjects with MDP or MDP/dog allergen did not modify their cytokine production. The lack of effect of MDP alone on human DC has been previously reported in other studies [45,46], whereas, to our knowledge, allergen co-stimulation has not been assessed previously on human cells. Nonetheless, such studies were performed in mice. In particular, intraperitoneal MDP administration, together with LPS-free OVA antigen, resulted in a Th2 profile by splenic cells restimulated in vitro with OVA [47] but not in a Th17 profile [26]. The difference with our study may be related to the species or to the absence of LPS in the OVA antigen, as compared with dog allergen, during the sensitization phase. When looking at the effects of stimulated DC from NA donors on the priming of naive T cells, no effects were observed on Th2, Th1, and bona fide classical Th17 cytokines IL-17A and IL-22. However, dog-stimulated DC-primed T cells exhibited an almost significant increase in IL-17F. IL-6, together with other cytokines, is needed for human Th17 differentiation [48,49], and, therefore, its production by dog-allergen-stimulated DC may play a role in the induction of IL-17F in naive T cells. Noteworthy, although IL-17F belongs to the Th17 lineage [50], and in sharp contrast with the deleterious asthma outcomes of IL-17A, it has a protective effect on experimental asthma development [51]. Altogether, these data, based on samples obtained from NA donors, confirm the animal model findings of a lack of a co-stimulatory effect between dog allergen and the NOD2 pathway, in the development of a deleterious Th2/Th17 response in nonsensitized subjects. They suggest that anti-infectious vaccines, using NOD2 ligands as adjuvants, might be relatively safe in children who have not yet developed asthma, although further studies are needed to evaluate this point. However, as NOD2 ligands are being associated with other PRR agonists in some vaccines, in particular TLR7 [20] and TLR2 [52] ligands, it remains to be established if such associations may not have other outcomes. Studies are quite reassuring for the TLR7/8 ligand, as, in contrast to NOD1, co-stimulation with the NOD2 agonist MDP does not induce the production of inflammatory cytokines by human DC and does not favor IL17 production by T cells [53]. It is less clear for TLR2/NOD2 association, which has additive effects on the production of IL-1β and IL-6 by human DC, but was not evaluated on the polarization profile of T cells [52]. In another study, TLR2/NOD2 co-activation induced the production of IL-1 β and IL-23 by human DC, which induced the production of IL-17 by memory but not naive T cells [24]. To next assess if NOD2 co-stimulation may alter the cytokine profile in already-allergic asthma patients, the same experiments were performed using DC obtained from dog-allergen-sensitized asthma patients. As observed with DC from NA subjects, dog-allergen stimulation only induced the production of IL-6 in DC and of IL-17F in T cell co-cultures. In contrast, both IL-1β and IL-6 were increased in DC supernatants after NOD2 co-stimulation, compared with MDP alone. Besides IL-6, IL-1β is also among the factors inducing the differentiation of Th17 cells [48,49], which has also been shown to promote Th2 responses in murine models of asthma [54,55]. Accordingly, increases in Th17 and Th2 cytokines were found in the co-cultures when DC were derived from AA patients, although only in the co-stimulated condition, as compared with MDP. These data suggest that DC from AA patients are imprinted in vivo and are able to promote both a Th2 and a Th17 profile, when co-triggered through the NOD2 pathway. In particular, DC from AA patients produced significantly more TGF-β, a cytokine shown to be mandatory for the development of a Th17 profile, than DC from NA subjects [49]. Altogether, these data highlight the differences between nonallergic subjects and allergic asthmatic patients, in their response to allergen/NOD2 ligand co-stimulation. The data suggest that infections in children not sensitized to airway allergens may not participate in the development of asthma through the NOD2 signaling circuitry. Care has to be taken in already-sensitized asthma patients, for vaccines using NOD2 ligands as adjuvant.

## 4. Materials and Methods

### 4.1. Mice

C57BL/6J female mice (6 weeks old) were purchased from Charles River or Janvier Lab (France). All animals were housed under specific pathogen-free conditions, in ventilated cages with absorbent bedding material, maintained on a 12 h daylight cycle and with free access to commercial pelleted food and water ad libitum.

### 4.2. Allergen Sensitization and Challenge

After isoflurane gas anesthesia, positive groups were sensitized i.n. with dog allergen extract (kindly provided by Stallergenes and ALK France), at a dose corresponding to 10 µg/mL of the major allergen Can f 1, alone or in combination with the NOD2 agonist MDP (InvivoGen), at 10 µg in 30 µL phosphate buffer saline (PBS), or, for the control groups, with PBS or MDP alone for 5 days. The dose of MDP was selected following the previous in vitro [40] and in vivo dose-response experiments and the literature analysis [39]. Nine days later, mice were challenged i.n., with the same quantity of dog allergen (positive groups) or PBS (control groups), 5 days a week for 3 weeks. Twenty-four hours later, mice were anesthetized, assessed for AHR, and sacrificed by pentobarbital injection (Euthasol, Centravet, Amiens, France) (Figure 1a). For all experiments, BAL fluid was recovered, and blood samples were collected to obtain serum. Lungs were collected for RNA isolation, protein analysis, and histology or flow cytometry analysis.

### 4.3. Airway Responsiveness Measurement

Mice were anesthetized with 0.5 mg/kg medetomidine (Domitor; Pfizer, New York, NY, USA) and 5 mg/kg ketamine (Imalgene 1000; Merial, Lyon, France) and immediately intubated with an 18-gauge catheter, followed by mechanical ventilation using a FlexiVent (SCIREQ). Mice were exposed to nebulized PBS, followed by increasing concentrations of nebulized methacholine (0–100 mg/mL) (Sigma-Aldrich) using an ultrasonic nebulizer (Aeroneb, Aerogen, Ireland). Return to baseline resistance was ensured, prior to the administration of the next doses of methacholine. For each dose, 10 cycles of nebulization and measurements were performed. Only resistance values corresponding to coefficient of determination values > 0.95 were kept. The maximal values of measured resistances were calculated for each dose.

### 4.4. BAL Analysis

A total volume of 1 mL of ice-cold PBS was used to gently wash the lungs. Cells from the lavage fluid were counted and then recovered by centrifugation at 1200 rpm for 5 min at 4 °C. Total leukocytes were then resuspended in PBS, to have 100,000 cells/0.1 mL, and resuspended BAL was cytocentrifuged (Shandon Cytospin 4; Thermo Fisher Scientific, Waltham, MA, USA) and stained with May-Grünwald Giemsa (DiaPath, Martinengo, Italy) for differential cell count.

### 4.5. Serum and BAL Antibodies Measurement

Blood was drawn from the upper cave vein, and serum was collected by centrifugation at 13,000 rpm for 1 min. Total IgE serum level was measured by ELISA, as previously described [40]. For dog-specific IgE antibodies in serum, 96-well plates (Corning Incorporated, New York, NY, USA) were coated with 10 µg/mL Can f 1-containing dog allergen in PBS. After blocking with 0.05% tween 5% milk in PBS, and serum or BAL samples addition, an anti-mouse IgE antibody coupled to HRP (Gentaur, Paris, France) was added in CGS2 buffer (Can Get Signal for secondary Antibody, Cosmo Bio, Japan), followed by addition of TMB substrate solution (Sigma Aldrich, Saint-Quentin Fallavier, France). For dog-specific IgG1 antibodies in BAL, 96-well plates were coated with 10 µg/mL Can f 1-containing dog allergen in PBS. After blocking with 3% BSA, and serum or BAL samples addition, an anti-mouse IgG1 antibody coupled to HRP (Southern Biotech, Birmingham, AL, USA) was added with 0.5% BSA in PBS, followed by addition of TMB substrate solution (Sigma Aldrich). For each ELISA, the OD value at 450 nm was determined. The dog-specific antibody titers of the samples were determined with a pooled standard that was generated in the laboratory. Titers were expressed as the inverse of the dilution, corresponding to 50% of the maximal optical density. Total IgA in BAL was assessed using an IgA mouse ELISA kit, in accordance with the instructions of the manufacturer (eBioscience, San Diego, CA, USA).

### 4.6. RNA Isolation and Quantitative RT-PCR

RNA was extracted from the lung using the NucleoSpin RNA mini kit (Macherey-Nagel, Hoerdt, France), in accordance with the instructions of the manufacturer. Extracted RNA was reverse-transcribed with the High-Capacity cDNA reverse transcription kit (Applied Biosystems, USA), in accordance with the instructions of the manufacturer. The resulting cDNA was amplified using the IDT Prime Time Assay master mix. Real-time PCR was performed with Prime-Time Probe Assay (IDT, Leuven, Belgium). *Il-13* (Mm.PT.58.31366752), *Il-4* (Mm.PT.58.7882098), *Il-17a* (Mm.PT.58.6531092), *IL-22* (mo IL-22 20×), and *Ifn-γ* (Mm.PT. 58.41769240). They were detected on a QuantStudio 12K Flex Real-Time PCR System (Applied Biosystems, Waltham, USA). Data were analyzed via the Thermo Fisher cloud. *Rplp0* (Ribosomal Protein Lateral Stalk Subunit P0) (Mm.PT.58.43894205) was used as internal reference gene to normalize the transcript levels. Relative mRNA level (2^−∆∆Ct^) was determined by comparing the PCR cycle thresholds (Ct) for the gene of interest and internal reference gene (∆Ct) and ΔCt values for treated and control groups (ΔΔCt).

### 4.7. Lung Protein Extracts

Lung protein extracts were prepared after mechanical dissociation (Precellys, Bertin Technologies SAS, Montigny-le-Bretonneux, France) of one lung lobe in 1 mL of lysis buffer (Tissue-Protein Extraction Reagents, Life Technologies, Villebon-sur-Yvette, France) and protease inhibitor cocktail (Roche, Newburyport, MA, USA) at 4 °C. Supernatants were collected for further total protein (Pierce BCA proteins Assay kit, Life Technologies), cytokine, and chemokine measurements.

### 4.8. Flow Cytometry

Lungs were perfused through the pulmonary artery and digested in 1 mg/mL type IV collagenase (Gibco, Life Technologies) and 2 IU/mL DNase I (Roche Diagnostics, Meylan, France) in RPMI 1640 (Dutscher, Bernolsheim, France) for 45 min at 37 °C. Cells were collected, filtered on a 70 μm pore membrane, and centrifuged at 670× *g* for 15 min on a Percoll density gradient (GE Heathcare, Chicago, IL, USA). Erythrocytes were lysed in Red Blood Cell Lysis Buffer (Biolegend, London, UK). Cell suspensions were stimulated for 2 h at 37 °C with 10 ng/mL phorbol 12-myristate 13-acetate (Invivogen), 0.5 µg/mL ionomycin (InvivoGen, Toulouse, France), and brefeldin A (eBioscience). Cells were then blocked with Fc Block (anti-CD16/CD32 antibody, BD-Biosciences, Le Pont de Claix, France). For cell surface staining, specific antibodies or corresponding control isotypes were added for 20 min. γδ cells were stained using CD3-AF700 and γδTCR-PE-Cy7 and were defined as CD3^+^TCRγδ^+^ cells. For T cells, CD45^+^ cell was selected with a CD45-APC-Cy7. The T cell population of interest was selected using CD3-AF700 and CD4-BV605 and was defined as CD4^+^: CD45^+^ CD3^+^ CD4^+^. Dead cells were excluded using LIVE/DEAD aqua dead stain (Thermofischer scientific, les Ulis, France). Cells were fixed and permeabilized using the BD Cytofix/Cytoperm kit and stained intracellularly with an anti-IL-17-BV421 antibody or the corresponding FMO or control isotypes. Details about antibodies are provided in Table 1. Cell fluorescence was measured by flow cytometry (LSR FORTESSA, BD Biosciences) and analyzed using Flow Jo v10.6.1. software. The gating strategy is presented in Figure 9.

### 4.9. Histology and Immunohistochemistry

The left unwashed lung from each mouse was fixed in Antigenfix (Microm Microtech, Brignais, France) and embedded in paraffin (Histowax, Microm Microtech), in accordance with the indications of the manufacturer. Lung sections of 5 μm (Microm HM355S Thermoscientific) were stained, and bronchial cell infiltration was quantified using the following score [56]: 0, no mononuclear cell infiltration; 1, few mononuclear cells; 2, from 1 to 5 layers of inflammatory cells around bronchi; 3, thickness of inflammatory cells > 5 cells and surrounding the entire bronchus. For airway remodeling, mucus production, collagen deposition, and smooth muscle cell immunostaining were assessed. For mucus, periodic acid–Schiff (PAS) staining kit (Clinisciences, Nanterre, France) was used for mucopolysaccharide staining. Mucus production in the airway epithelium was quantified based on a five-point system: 0, no mucus; 1, <25% of the epithelium; 2, 25%–50% of the epithelium; 3, 50%–75% of the epithelium; 4, >75% of the epithelium, as previously described [57]. For collagen deposition, sections were stained with Masson’s trichrome (Clinisciences). Quantification of peribronchial collagen deposition by Masson’s trichrome was scored as [58]: 1, marginal peribronchial trichrome stain; 2, slight increase in peribronchial trichrome stain; 3, increased peribronchial thickness in trichrome stain; 4, dramatically increased thickness of trichrome stain in all airways. Data were expressed as mean scores per bronchus. Smooth muscle cells were identified by immunohistochemistry using a primary rabbit anti-mouse α-SMA antibody (clone E184, Abcam, Paris, France) and the rabbit Vectastain ABC-Alkaline Phosphatase kit (Vector Laboratories, Peterborough, UK) and were developed using fast red (Sigma Aldrich). Quantification was performed in small bronchi using image J software and was expressed as µm^2^ surface area/µm of bronchus basal membrane.

For iBALT analysis, the number and surface area of iBALT follicles were counted. iBALT was defined as lymphoid aggregates, composed of B cell follicles interconnected with FDC and surrounded by T cells, situated adjacent to a bronchus and next to a vein and an artery [59,60]. Therefore, isolated parenchymal or perivascular follicles were not counted. Data were expressed as number of iBALT per bronchus, and surface area as number of squares (each 0.025 mm^2^) counted with an ocular grid per iBALT.

iBALT B cells, T cells, and FDC were identified by immunohistochemistry. Sections were incubated with a primary rabbit monoclonal antibody against murine CD21 FDC (Sigma Aldrich, clone EP3093), rat antibody against CD45R B cells (clone RA3-6B2, eBioscience), and a rabbit antibody against CD3 T cells (clone SP7, Abcam), after a step of antigen retrieval. The corresponding Vectastain ABC-AP kit was then used, and the color was developed using Fast Red substrate kit (Sigma Aldrich). Numbers of positive cells were enumerated using an ocular grid and were expressed as number of positive cells per iBALT and per iBALT surface area (number of grid squares).

For all data, measurements were performed in 3 to 9 different lung sections and were averaged for each mouse. Quantification was performed blindly in whole lung sections. Images were acquired on a DM 3000 LED (Leica, Wetzlar, Germany) light microscope.

### 4.10. Human Donors

Venous blood was obtained from 10 healthy NA subjects, with no history of allergic diseases, exhibiting total immunoglobulin E (IgE) levels < 100 kU/L and absence of allergen-specific IgE antibodies (Etablissement français du sang). Venous blood was also collected from 10 AA patients. These patients exhibited a history of allergic asthma to dog allergen, positive allergen-specific IgE antibodies (>3 kU/L), and elevated total IgE levels. None had received oral corticosteroids within 2 months before sample collection, while 7/10 were under inhaled corticosteroids. Characteristics of the asthma patients are shown in Table 2.

### 4.11. Generation of DC

Peripheral blood mononuclear cells (PBMCs) were isolated by density gradient centrifugation using Ficoll-Hypaque (GE Healthcare, Chicago, IL, USA). CD14^+^ monocytes were isolated by magnetic column purification, on the basis of positive selection, with anti-CD14 microbeads (Miltenyi Biotec, Bergisch Gladbach, Germany) with a purity ≥ 96%. To generate DC, 1 × 10^6^ monocytes were cultured in complete RPMI 1640 medium supplemented with granulocyte-macrophage colony-stimulating factor at 25 ng/mL (Miltenyi Biotec) and IL-4 at 10 ng/mL (Peprotech, Cranbury, NJ, USA) for 6 days, as previously described [61]. DC were then matured for 48 h with either 10 µg/mL Can f 1-containing dog allergen, NOD2 ligand (MDP) at 10 μg/mL (Invivogen CAYLA), or a combination of dog allergen and MDP. Endotoxin concentration was less than 0.05 EU/mL in MDP, and 83.3 ng/mL in dog allergen, as assessed by the chromogenic Limulus Amoebocyte Lysate assay (Lanza). Supernatants were collected and stored at −20 °C for further cytokine evaluation. Cells were harvested and used for T cell co-cultures.

### 4.12. Isolation of CD4^+^CD45RA^+^ Naive T Cells and DC/T Cell Co-Cultures

Allogeneic CD4^+^CD45RA^+^ naive T cells were obtained from PBMCs exclusively from healthy donors, by negative selection using naive CD4^+^ T Cell Isolation Kit II (Miltenyi Biotec), as previously described [62], with a purity ≥ 95%. Mature DC were co-cultured with naive T cells, at a ratio of 1:10 at 10^6^ cells/mL in complete RPMI for 5 days. Supernatants were recovered and frozen at −20 °C until use.

### 4.13. Cytokine and Chemokine Protein Assessment

For murine samples, the levels of IL-33 cytokine and CCL17, CXCL1, and CXCL2 chemokines were assessed using commercial ELISA kits (detection limits of 25, 31.2, and 15,6 pg/mL, respectively), in accordance with the instructions provided by the manufacturers (eBioscience, and R&D Systems, Minneapolis, MN, USA). Data are expressed as pg per mg of total proteins. For human cells, concentrations of IL-1β, IL-6, TGF-β1 (R&D Systems), and IL-23p19 (eBioscience) in DC supernatants and of IL-17A, IL-22, IL-13, IL-5 (R&D Systems), IL-17F (eBioscience), and IFN-γ (BD Bioscience) in DC/T-cell co-culture supernatants were measured by ELISA, in accordance with the recommendations of the manufacturer, and expressed as ng/mL.

### 4.14. Statistical Analysis

Normally distributed data (Shapiro–Wilk normality test) were analyzed by one-way analysis of variance (ANOVA), followed by post hoc multiple comparison test (Sidak test). Non-normally distributed data were analyzed by the Kruskal–Wallis test, followed by multiple comparison tests. For AHR, a two-way ANOVA test was used. For paired human in vitro experiments, data were analyzed using the Friedman test. GraphPad Prism v9 software was used. *p* values < 0.05 were considered statistically significant.

## Figures and Tables

**Figure 1 ijms-23-11894-f001:**
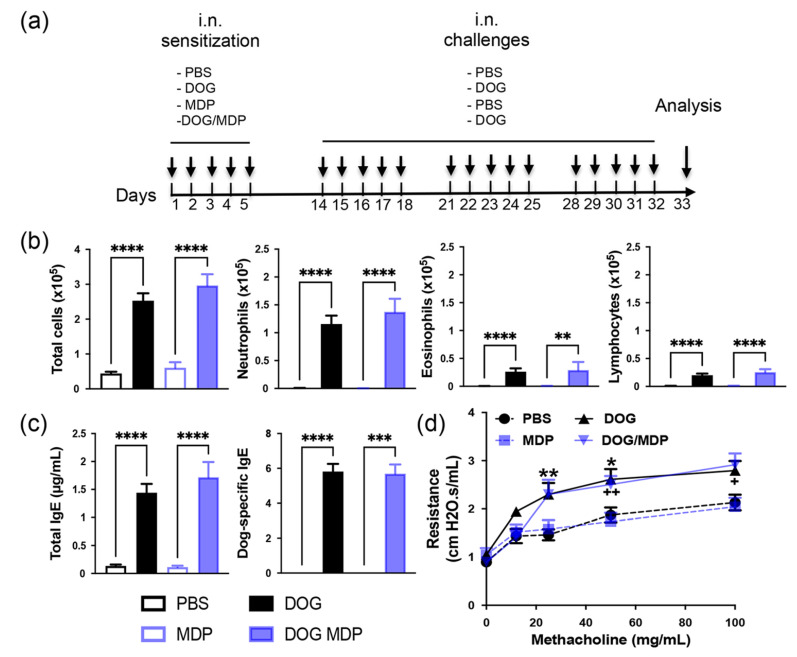
NOD2 ligand co-administration does not aggravate allergen-induced airway cell recruitment, humoral response, and AHR. (**a**) Protocol of dog-induced experimental AAI, with or without NOD2 agonist co-administration. (**b**) BAL cell counts in control (PBS and MDP) and challenged (dog and dog/MDP) mice. (**c**) ELISA measurement of total IgE and dog-specific IgE (as titers) in sera. (**d**) AHR in control and challenged mice. Data are presented as mean ± SEM (n = 7–22 mice per group, or n = 5–11 mice per group for AHR). * *p* < 0.05, ** *p* < 0.01, *** *p* < 0.001, **** *p* < 0.0001 versus PBS or indicated column; + *p* < 0.05, ++ *p* < 0.01 versus MDP. Kruskal–Wallis test for all, except AHR, which is 2-way ANOVA.

**Figure 2 ijms-23-11894-f002:**
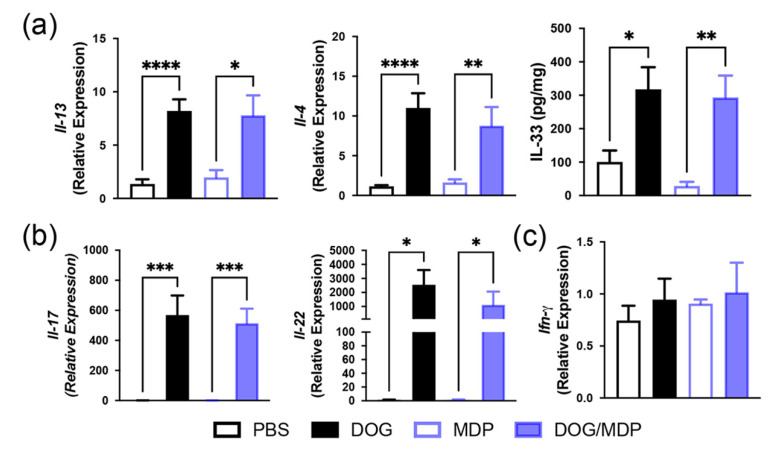
The mixed Th2/Th17-type profile induced by dog allergen remains unchanged under the effect of NOD2 ligand. Relative mRNA expression and protein level of (**a**) Th2 cytokines, (**b**) Th17 cytokines, and (**c**) Th1 cytokine. Cytokines were assessed by quantitative real-time PCR and by ELISA in lung tissues from control and challenged mice. Data are presented as mean ± SEM (n = 5–20 mice per group). * *p* < 0.05, ** *p* < 0.01, *** *p* < 0.001, **** *p* < 0.0001 versus PBS or MDP. Kruskal–Wallis test for all.

**Figure 3 ijms-23-11894-f003:**
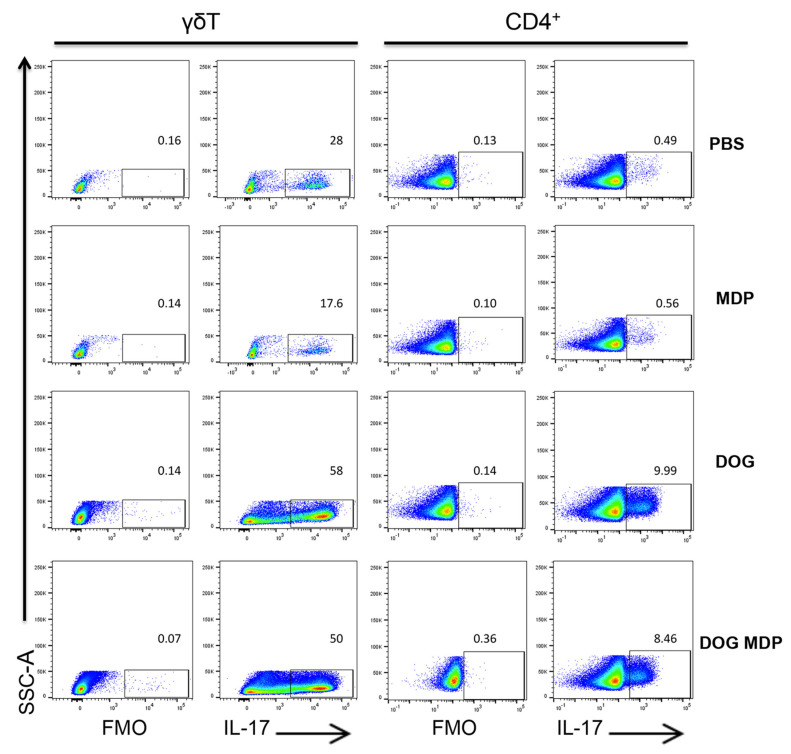
Flow cytometry analysis of IL-17 expression in lung CD4^+^ and γδ T cells. Cells were gated as described in the methods section. Percentage of cytokine positive cells among each cell population is shown above the gate. One representative experiment out of three.

**Figure 4 ijms-23-11894-f004:**
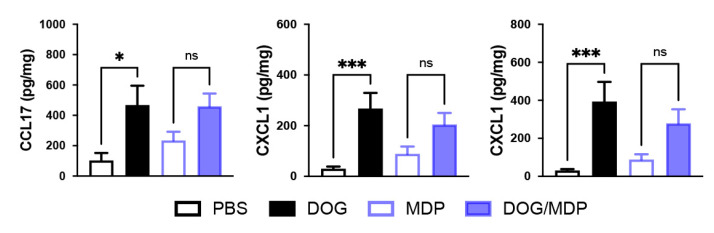
NOD2 ligand co-stimulation does not modify allergen-induced lung chemokine production. Protein levels of CCL17, CXCL1, and CXCL2. Chemokines were assessed by ELISA in lung tissues from control and challenged mice. Data are presented as mean ± SEM (n = 4–18 mice per group). * *p* < 0.05, *** *p* < 0.001 versus PBS, ns: not significant. Kruskal–Wallis test for all.

**Figure 5 ijms-23-11894-f005:**
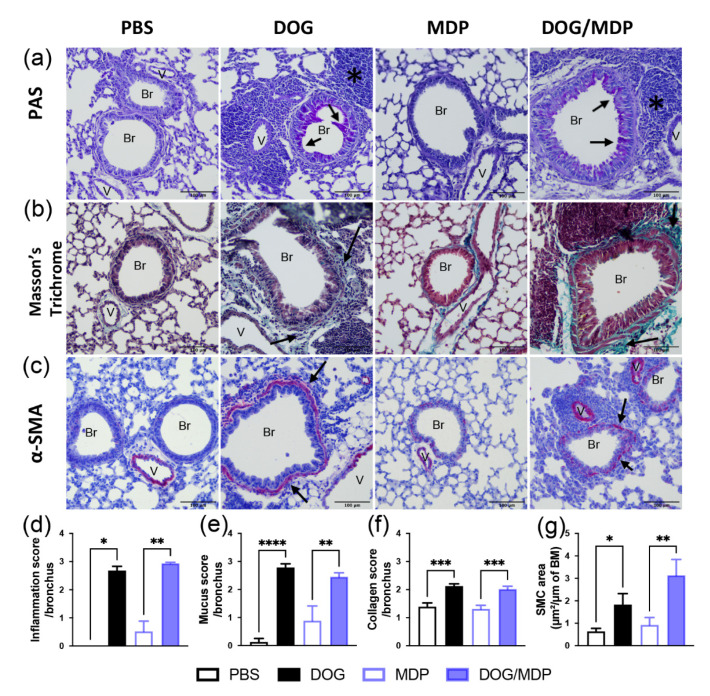
Dog-allergen-induced AAI is associated with bronchial remodeling without additional effect of MDP. Representative microphotographs and quantification of stained paraffin-embedded lung sections for (**a**) inflammation and mucus production using PAS (pink color), (**b**) collagen using Masson’s trichrome (blue/green staining), and (**c**) immunohistochemistry staining of SMC using α-smooth muscle actin (SMA) antibody (positive cells are stained in red). Scale bars: 100 μm. Black arrows show staining of interest (mucus, collagen, and SMC), and black stars show inflammation. V: vessel, Br: bronchus. Quantification of lung sections for (**d**) inflammation, (**e**) mucus, (**f**) collagen, and (**g**) SMC staining. Data are presented as mean ± SEM (n = 4–7 mice per group, with 3 to 9 different lung sections analyzed). * *p* < 0.05, ** *p* < 0.01, *** *p* < 0.001, **** *p* < 0.0001 versus PBS or MDP. Kruskal–Wallis test for all, except for figures (**e**) and (**f**), which are 1-way ANOVA.

**Figure 6 ijms-23-11894-f006:**
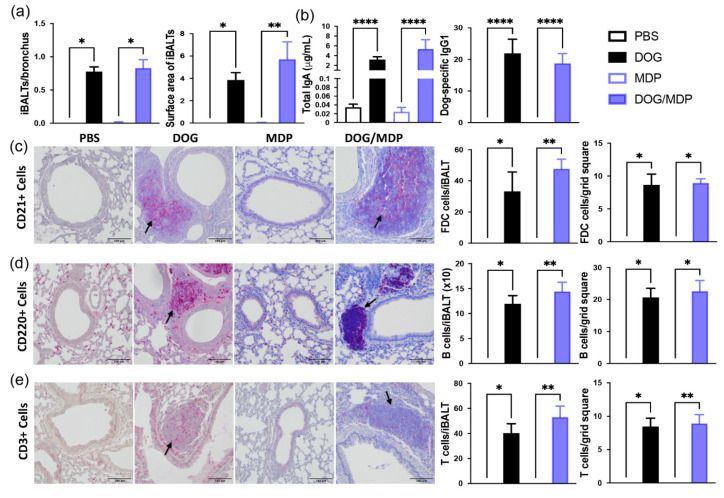
MDP does not aggravate the formation of iBALT. Representative microphotographs and quantification of iBALT. (**a**) Numbers and surface area (number of grid squares) of iBALT. (**b**) Levels of total IgA and dog-specific IgG1 (as titers) in BAL. (**c**–**e**) Immunohistochemistry staining of FDC (**c**), B (**d**), and T (**e**) cells and counts and surface area per iBALT. Scale bars: 100 μm. Positive cells are stained in red. Black arrows show staining of interest. Data are presented as mean ± SEM (n = 4–6 mice per group, with 3 to 9 different lung sections analyzed per mouse, for histology, and n = 9–18 mice per group for Ig). * *p* < 0.05, ** *p* < 0.01, **** *p* < 0.0001 versus PBS or MDP. Kruskal–Wallis test for all.

**Figure 7 ijms-23-11894-f007:**
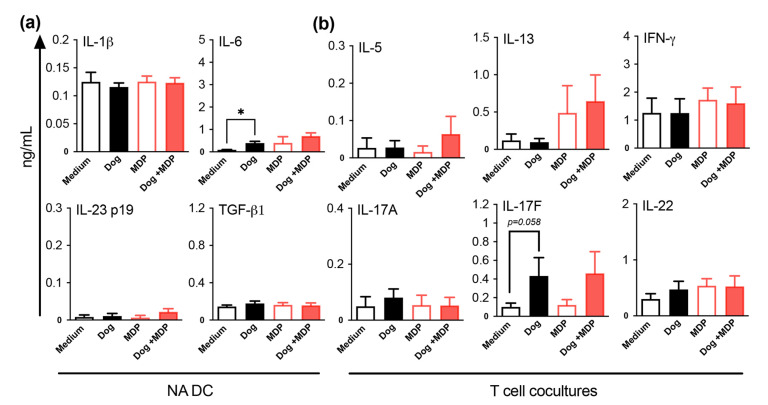
MDP co-stimulation of DC from NA subjects does not alter DC cytokine production and T cell polarization. (**a**) Cytokine profile of DC supernatants quantified by ELISA. (**b**) Cytokine profile of DC/T cell supernatants evaluated after 5 days of co-culture by ELISA. Data are presented as mean ng/mL ± SEM for n = 8–10 NA subjects. * *p* < 0.05 versus medium. Friedman test for all.

**Figure 8 ijms-23-11894-f008:**
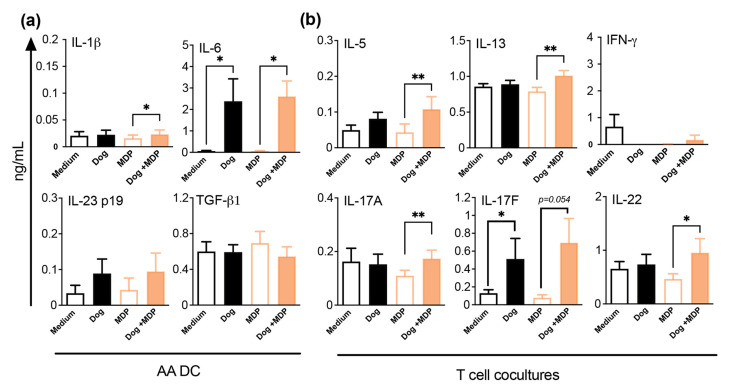
MDP co-stimulation of DC from AA subjects promotes DC cytokine production and Th2/Th17 cell polarization. (**a**) Cytokine profile of DC supernatants quantified by ELISA. (**b**) Cytokine profile of DC/T cell supernatants evaluated after 5 days of co-culture by ELISA. Data are presented as mean ng/mL ± SEM for n = 10 AA. * *p* < 0.05 versus medium or MDP, ** *p* < 0.01 versus MDP. Friedman test for all.

**Figure 9 ijms-23-11894-f009:**
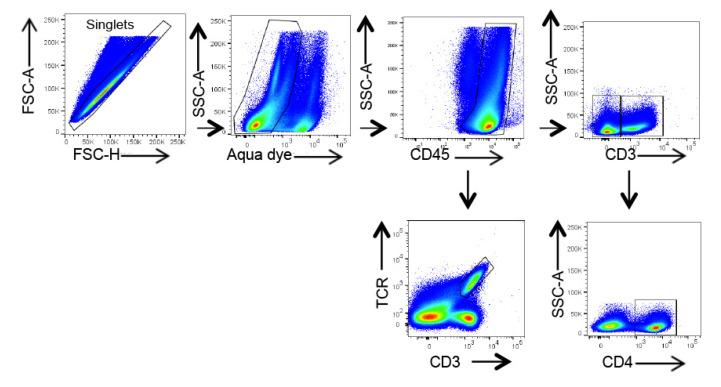
Gating strategy of γδ and CD4^+^ T cell subsets in the lung. Gating strategy included forward and side scatter gating, exclusion of multiplets, selection of living cells with aqua dead stain, and selection of cell subsets according to the indicated cell surface markers.

**Table 1 ijms-23-11894-t001:** Antibodies used for flow cytometry assays.

Antibody	Catalog Number	Provider	Quantity (µg)
TCRγδ PE-Cy7	25-5711-82	Invitrogen	0.3
CD3 Alexa Fluor 700	561388	BD	0.3
CD4 Brilliant Violet 605	100547	Biolegend	0.25
CD45 APC Cy7	557659	BD	0.3
IL-17A Brilliant Violet 421	506925	Biolegend	0.25

**Table 2 ijms-23-11894-t002:** Asthma patients’ characteristics.

Sex (F/M)	8/2
Age (y)	37.42 ± 3.36
FEV1 (% predicted)	88.13 ± 3.16
FEV1/FVC ratio (%)	74.14 ± 3.27
Specific IgE (KU/L)	10.33 ± 4.74
μg Inhaled CS (equivalent beclometasone dipropionate)	614.6 ± 215
Smokers (n)	2/10
BMI above 25 (n)	1/10
ACT ≥ 20 (%)	80
Intermittent (n)	3/10
Moderate (n)	5/10
Severe (n)	2/10

Data are presented as n or mean ± SEM, unless otherwise stated. Y: year; FEV1: forced expiratory volume in 1 s; FVC: forced vital capacity; CS: cortico steroids; BMI: body mass index (kg/m^2^); ACT asthma control test (patients are controlled when ≥ 20); intermittent asthma is defined as requiring only B2 agonists as needed, moderate asthma as requiring less than 1000 μg inhaled CS, and severe asthma as requiring high doses of CS with add-on therapy.

## Data Availability

The original contributions presented in the study are included in the article’s materials. Further inquiries can be directed to the corresponding author.

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
