# Peer review of "NOD2 Signaling Circuitry during Allergen Sensitization Does Not Worsen Experimental Neutrophilic Asthma but Promotes a Th2/Th17 Profile in Asthma Patients but Not Healthy Subjects"

_ijms, 2022, doi:10.3390/ijms231911894_

Round 1

Reviewer 1 Report

In this study, Bouté et al. investigated the effect of nucleotide-binding oligomerization domain 2 (NOD2) stimulation during allergen sensitization. In general, the study is appropriately designed, and the manuscript is very well written. The findings are comprehensive and interesting. However, there are a few minor questions that have to be answered in order for the manuscript to be published in this journal. 

Minor comments:

1. The authors haven’t provided sufficient evidence to show that a desirable amount of the muramyl di-peptide (MDP) reached the target cells in the lung in order to stimulate NOD2 and activate the downstream signalling cascade.

2. In the method section, the authors have not stated which device was used for airway responsiveness measurements in mice.

3. If possible, I would suggest shortening the title. 

Author Response

Reviewer 1

In this study, Bouté et al. investigated the effect of nucleotide-binding oligomerization domain 2 (NOD2) stimulation during allergen sensitization. In general, the study is appropriately designed, and the manuscript is very well written. The findings are comprehensive and interesting. However, there are a few minor questions that have to be answered in order for the manuscript to be published in this journal.  

Minor comments: 

  1. The authors haven’t provided sufficient evidence to show that a desirable amount of the muramyl di-peptide (MDP) reached the target cells in the lung in order to stimulate NOD2 and activate the downstream signalling cascade. 

Answer: We had previously shown that the selected MDP dose was able to activate bronchial epithelial cells inducing cytokine release (ref 40). We also performed in vivo dose-response experiments with various quantities of MDP that did not show any additional effect for doses higher than the one used. We finally chose the same intra-nasal dose as the study from Duan et al (ref 39), which exhibited changes in their model of tolerance in asthma. Furthermore, we have confirmed that MDP administered via the intra-nasal route reaches all lobes of the lung by performing an administration of Evans blue dye. This information has been added to the materials and methods (page 12).

  1. In the method section, the authors have not stated which device was used for airway responsiveness measurements in mice.

Answer : Sorry, this information was inadvertenly deleted. Airway responsiveness was assessed using the FlexiVent (SCIREQ ®), and this information has been added to the materials and methods (page 13).

  1. If possible, I would suggest shortening the title. 

As asked by the reviewer, we have shortened the title : NOD2 signaling circuitry during allergen sensitization does not worsen experimental neutrophilic asthma, but promotes a Th2/Th17 profile in asthma patients but not healthy subjects.

Reviewer 2 Report

The authors have attempted to determine whether muramyl di-peptide (MDP) given as an adjuvant during allergen sensitization may worsen the development of Th2/Th17 responses. The authors found that intranasal co-administration of MDP did not modify the AAI parameters including Th2/Th17 type lung inflammation in a murine model of airways inflammation. In vitro, MDP co-stimulation of aller-24 gen-primed DC did not change the polarization profile of T cells in healthy subjects but elicited a Th2/Th17 profile in asthma subjects. Taken together, the data suggest that NOD2 adjuvants might be used in non-sensitized individuals but should be careful in usage in asthmatics. The study was well designed and presented logically.

This reviewer only has only some minor comments:

  1. It has been noted that C57BL/6j mice had been used in the present study. I wonder if this type of mice might affect the phenotype of asthma-like changes in these mice compared with BALB/c mice.
  2. In the methods, the authors mentioned that total and specific IgE were also measured in the animal models. However, there seem no relevant data shown.  
  3. It could be better if the authors provide general information of subjects who donated clinical samples in a table.

Author Response

Reviewer 2

The authors have attempted to determine whether muramyl di-peptide (MDP) given as an adjuvant during allergen sensitization may worsen the development of Th2/Th17 responses. The authors found that intranasal co-administration of MDP did not modify the AAI parameters including Th2/Th17 type lung inflammation in a murine model of airways inflammation. In vitro, MDP co-stimulation of allergen-primed DC did not change the polarization profile of T cells in healthy subjects but elicited a Th2/Th17 profile in asthma subjects. Taken together, the data suggest that NOD2 adjuvants might be used in non-sensitized individuals but should be careful in usage in asthmatics. The study was well designed and presented logically. 

This reviewer only has only some minor comments:

  1. It has been noted that C57BL/6j mice had been used in the present study. I wonder if this type of mice might affect the phenotype of asthma-like changes in these mice compared with BALB/c mice.

Answer: It is known that C57BL6 mice are more prone to develop Th1 responses whereas BALB/c mice develop preferentially Th2 responses. However as most of the genetically modified mice are developed on a C57BL6 background, many groups use this genetic background. We have used both mouse backgrounds, achieving similar results, even though the C57BL6 background required higher doses of allergen. Altogether, we are able to obtain a good asthma phenotype using this mouse background.

2. In the methods, the authors mentioned that total and specific IgE were also measured in the animal models. However, there seem no relevant data shown.  

Answer : The results for total and specific IgE are shown in figure 1c (page 3), and there were no differences between dog and dog/MDP challenged mice.

3. It could be better if the authors provide general information of subjects who donated clinical samples in a table.

Answer: As suggested by the reviewer, we have added a table recapitulating patients general information  (table 3 page 16).